# microRNA Targeting Cytochrome P450 Is Involved in Chlorfenapyr Tolerance in the Silkworm, *Bombyx mori* (Lepidoptera: Bombycidae)

**DOI:** 10.3390/insects16050515

**Published:** 2025-05-12

**Authors:** Ying Shao, Jian-Hao Ding, Wang-Long Miao, Yi-Ren Wang, Miao-Miao Pei, Sheng Sheng, Zhong-Zheng Gui

**Affiliations:** 1Jiangsu Key Laboratory of Sericultural and Animal Biotechnology, School of Biotechnology, Jiangsu University of Science and Technology, Zhenjiang 212100, China; shaoyingfs@163.com (Y.S.); lostone1@126.com (J.-H.D.); 18390849036@163.com (W.-L.M.); 15387693229@163.com (Y.-R.W.); 18943432267@163.com (M.-M.P.); parasitoids@163.com (S.S.); 2Key Laboratory of Silkworm and Mulberry Genetic Improvement, Ministry of Agriculture and Rural Affairs, The Sericultural Research Institute, Chinese Academy of Agricultural Sciences, Zhenjiang 212100, China

**Keywords:** *Bombyx mori*, chlorfenapyr, insecticide tolerance, P450s, microRNA

## Abstract

Silkworm *Bombyx mori* larvae exhibit higher tolerance to chlorenapyr compared to other insecticides. However, the details of the associated tolerance mechanism remain poorly understood. In this study, we confirmed that chlorfenapyr can be biotransformed into tralopyril in the larvae bodies by HPLC analysis. Then, a differential transcriptomic database of sRNA was constructed based on chlorfenapyr-treated and healthy *B. mori* larvae, and a total of 342 known miRNAs and 83 novel miRNAs were obtained. Ten differentially expressed miRNAs were screened. Bmo-miR-6497-5p was significantly upregulated in the third instar silkworm larvae after treatment with chlorfenapyr, and its target CYP450 gene *CYP337A2* was downregulated. In addition, the luciferase activity significantly decreased after Bmo-miR-6497-5p bound to *CYP337A2*. The bioassay revealed that the mortality of silkworm larvae injected with the antagomir of Bmo-miR-6497-5p was significantly increased after exposure to sublethal concentration chlorfenapyr. It is possible that Bmo-miR-6497-5p may target *CYP337A2*, regulating its expression, and silkworm increases its chlorfenapyr tolerance by upregulating Bmo-miR-6497-5p expression, thereby inhibiting the biotransformation of chlorfenapyr to toxic tralopyril in a manner catalyzed by CYP337A2. The present study reveals the role of microRNA in the tolerance to chlorfenapyr in silkworms, improving our understanding of insecticide resistance in Lepidopteran pests.

## 1. Introduction

The silkworm, *Bombyx mori*, has been domesticated as an important economic insect for thousands of years. Mulberry foliage is the exclusive food resource for the silkworm. Notably, it can also be digested by multiple insect pests, presenting an extensive threat to sericulture [1,2]. To manage these pests, a mass of chemical insecticides is applied in mulberry fields; however, these insecticides can also have negative consequences for the survival and growth of the silkworm [3]. Meanwhile, long-term application and unreasonable use of these chemicals have led to increasingly serious resistance in the pests [4]. Due to a lack of higher selectivity which is efficient for mulberry pests while maintaining safety for silkworms, many novel insecticides are still forbidden in the management of mulberry pests [5].

Chlorfenapyr [(4-bromo-2-(4-chlorophenyl)-1-ethoxymethyl-5-trifluoromethylpyrrole-3-carbonitrile)] is a novel N-substituted halogenated pyrrole compound and pro-insecticide, which can be activated by the oxidative in vivo removal of the N-ethoxymethyl group by monooxygenases, leading to interruption of the conversion of ADP to ATP in the mitochondria and eventually the death of the organism [2,6]. It well established that chlorfenapyr is transformed into toxic tralopyril catalyzed by P450s and other enzymes, which perform certain functions in this form after entering the haemocoel in insects [7,8,9,10]. Over the past few decades, chlorfenapyr has been demonstrated to possess excellent broad-spectrum acaricidal and insecticidal activity [4]. Interestingly, the previous studies have shown that silkworms have a much higher tolerance to chlorfenapyr when compared to other mulberry-fed insects [2,11]. To date, the specific biochemical and molecular mechanism underlying this tolerance divergence remains largely unknown.

It is well established that cytochrome P450s play an important physiological role in various important physiological activities of insects. A large number of studies have shown that P450s play key roles in the response to pesticide stress and the formation of pesticide resistance in insects. For example, CYP6G1 has been shown to mediate the metabolism of imidacloprid in *Drosophila melanogaster* [12], and in *Nilaparvata lugens*, *CYP6AY1* and *CYP6ER1* also facilitate the metabolization of imidacloprid [13]. *CYP6CM1* is of great importance in the metabolism of pymetrozine in the whitefly *Bemisia tabaci* [14]. In addition, insect P450s also participate in the biosynthesis and metabolism of endogenous compounds such as pheromones, 20-OH ecdysone, and juvenile hormone [15]. Therefore, in-depth analysis of the toxicological mechanism of cytochrome P450 in insecticide resistance or tolerance can be expected to contribute to more fully understanding the molecular mechanism of insect resistance and tolerance.

MicroRNAs are small non-coding RNAs that are 19–24 nt long and play important regulatory roles in various life activities in eukaryotes by targeting the 3′-untranslated regions (UTR) or 5′-UTR of mRNA that share sequences which complement the “seed” region (2–8 nt) of the miRNA [16]. P450 gene expression can be regulated both transcriptionally and posttranscriptionally. At the posttranscriptional level, microRNAs (miRNAs) with a length of approximately 22 nucleotides can affect the posttranscriptional regulation of P450 genes through binding to coding sequences, 3′ UTRs, or 5′ UTRs [17,18,19]. Recent studies have shown that miRNA mediates the resistance of pests to insecticides through targeting and regulating the expression of P450 genes. For example, miR-285 has been found to be upregulated in pyrethroid-resistant strains of *Culex pipiens pallens*, thereby reducing the expression level of its target P450 gene *CYP6N23* [20]. Therefore, whether the alternation of the expression of P450 genes in silkworm larvae treated with chlorfenapyr is affected by miRNA regulation and ultimately leads to the tolerance of chlorfenapyr is worthy of further investigation.

In this study, first, the sensitivity of silkworms to chlorfenapyr and tralopyril was determined, and the metabolism dynamic of chlorfenapyr in the bodies of silkworm larvae was measured. Then, the differentially expressed miRNAs in silkworms treated with chlorfenapyr were screened via small RNA transcriptome sequencing, and the target P450 genes of differentially expressed microRNAs were predicted. By means of dual luciferin reporter gene verification and miRNA mimics/inhibitor injection, whether miRNA-mediated inhibition of P450 expression was involved in the chlorfenapyr tolerance of silkworms was verified. This study provides an important understanding of the molecular mechanism underlying insecticide tolerance in silkworm, generating novel insights into the mechanism of pesticide resistance in insects.

## 2. Materials and Methods

### 2.1. Insects and Chemicals

The *B. mori* (JS strain) larvae were collected from the Sericultural Research Institute, Jiangsu University of Science and Technology and were maintained in an insectary [25 ± 1 °C, 60–80% relative humidity (RH), 14-/10-h (light/dark) photoperiod], feeding on fresh mulberry leaves. Chlorfenapyr (purity 97.5%) and tralopyril (purity 98.0%) were obtained from the Department of Entomology, Nanjing Agricultural University, Nanjing, China.

### 2.2. Bioassays

The third instar larvae of *B. mori* were collected and fed with chlorfenapyr or tralopyril using leaf-dipping bioassays. Briefly, the chlorfenapyr and tralopyril were diluted with distilled water containing 0.1% Triton X-100 to prepare the stock solution and to generate five serial dilutions (chlorfenapyr: 80.0, 160.0, 320.0, 640.0, 1280.0 mg/L; tralopyril: 1.0, 2.0, 4.0, 8.0, 16.0 mg/L), respectively, facilitating uniform spread of the active ingredient on the mulberry leaf surface. Leaf discs (7 cm diameter) were cut and dipped into the insecticide solutions for 30 s and air-dried for 30 min at room temperature. The discs were placed individually in plastic boxes and 30 third instar larvae were introduced per box. The mortality of the larvae was recorded after 48 h. Larvae were considered dead if they could not be induced to move when probed with a brush. Each concentration of chlorfenapyr or tralopyril was replicated three times, and each replicate involved 30 silkworm larvae. All bioassays were conducted under the following conditions: 25 ± 1 °C, 60–70% RH and 16:8 h (light/dark) photoperiod.

### 2.3. Analysis of Chlorfenapyr Biotransformation Using High-Performance Liquid Chromatography (HPLC)

In order to validate whether chlorfenapyr can be biotransformed into tralopyril and determine the metabolism dynamics of chlorfenapyr in the bodies of silkworm, HPLC was performed to determine the chlorfenapyr and tralopyril content at 0, 24, 48, and 72 h after the third instar larvae of *B. mori* were fed with mulberry leaves soaked with approximate LC_30_ chlorfenapyr. Once the larvae were collected, they were ground with liquid nitrogen, following which 500 μL of methanol was added for extraction. The mixture was centrifuged at 13,200 rpm for 10 min, and then the supernatant was collected and filtered using a 0.25 μm organic filter membrane. Chlorfenapyr and tralopyril were analyzed via Essentia LC-16 (Shimadzu, Kyoto, Japan) with a Kromasil C18 column (250 × 4.6 mm, 5 μm). The analytical conditions were as follows: the column temperature was set to 30 °C, the detection wavelength was 254 nm, the mobile phase was methanol/water (7:3, *v*/*v*), and the flow rate was 1.0 mL/min.

### 2.4. RNA Extraction and Library Construction

The third instar *B. mori* larvae were treated with the approximate LC_30_ chlorfenapyr or tralopyril, and 48 h later, total RNA was extracted using the mirVana miRNA Isolation Kit (Ambion, Sydney, Australia) according to the manufacturer’s protocol. Quantitation of total RNA was carried out using the Nanodrop 2000 (Thermo Fisher Scientific Inc., Waltham, MA, USA). RNA integrity was assessed with the Agilent 2100 Bioanalyzer (Agilent Technology, Santa Clara, CA, USA). One μg total RNA of each sample was used for the small RNA library construction using TruSeq Small RNA Sample Prep Kits (Cat. No. RS-200-0012, Illumina, San Diego, CA, USA) following the manufacturer’s recommendations. Briefly, total RNA was ligated to adapters at each end. Then, the adapter-ligated RNA was reverse transcribed to cDNA, and PCR amplification was performed. The PCR products ranging from 140 to 160 bp were isolated and purified as small RNA libraries. Library quality was assessed on the Agilent Bioanalyzer 2100 system using DNA High Sensitivity Chips. The libraries were finally sequenced using the Illumina HiSeq X Ten platform, and 150 bp paired-end reads were generated. The small RNA sequencing and analysis were conducted by OE Biotech Co., Ltd. (Shanghai, China).

### 2.5. Bioinformatic Analysis

The basic reads were converted into sequence data (also called raw data/reads) by base calling. Low quality reads were filtered, and the reads with 5′ primer contaminants and poly (A) were removed. The reads without the 3′ adapter and insert tag were filtered from the raw data, as well as those that were shorter than 15 nt or longer than 41 nt, and the clean reads were finally obtained. For primary analysis, the length distribution of the clean sequences in the reference genome was determined. The clean read sequences were aligned with Rfamdatabase (version 10.0, https://rfam.org/) (accessed on 20 September 2022) using the bowtie software, and rRNA, scRNA, Cis-reg, snRNA, tRNA, and other sequences were annotated and filtered. The known miRNAs were identified through alignment with the miRBase v22 database (http://www.mirbase.org/) (accessed on 20 September 2022), and the known miRNA expression patterns in different samples were analyzed. After that, unannotated reads were analyzed by mirdeep2 to predict novel miRNAs [21]. Based on the hairpin structure of a pre-miRNA and the miRBase database, the corresponding miRNA star sequence and miRNA mature sequence were also identified. Differentially expressed miRNAs were calculated and filtered with the threshold of *p* value < 0.05. The *p* value was calculated with the DEG algorithm in the R package for experiments with biological replicates and with Audic Claverie statistic for experiments without biological replicates [22]. The targets of differentially expressed miRNAs were predicted by using miRanda software (V0.3.0) [23], with the parameters as follows: S ≥ 150, ΔG ≤ −30 kcal/mol and demanding strict 5′ seed pairing. Gene Ontology (GO) enrichment and Kyoto Encyclopaedia of Genes and Genomes (KEGG) pathway enrichment analysis of different expressed miRNA-target-genes were performed using the R software (version: 4.0.0) based on the hypergeometric distribution.

### 2.6. Quantitative Real-Time PCR (qRT-PCR)

Total RNA was extracted using Trizol (Invitrogen, Waltham, MA, USA) according to the manufacturer’s instructions and treated with DNaseI. The concentration and purity of RNA samples were confirmed using a 2100 Bioanalyzer (Agilent Technologies, USA). The agarose gel electrophoresis was used to ensure the RNA integrity. The PrimeScript^®^ RT reagent Kit (Takara, Dalian, China) was used to produce first-strand DNA, followed by reverse transcription following the manufacturer’s recommendations. qRT-PCR was performed using QuantStudioTM 6 Flex (Thermo, Waltham, MA, USA) and ChamQ SYBR qPCR Master Mix (Vazyme, Nanjing, China). The primers used for qRT-PCR are listed in Appendix A. A cDNA dilution series (1, 1/3, 1/9, 1/27, and 1/81) with the sample cDNA was used to construct the standard curve and calculate the efficiency of amplification. A dissociation curve analysis was used to verify amplification of a single product. The qPCR conditions were as follows: 95 °C for 5 min; 40 cycles of 95 °C for 15 s; and 60 °C for 31 s. Reactions for all samples were run in triplicate with three biological replicates, and the relative expression level of each gene was calculated using the 2^−ΔΔCt^ method [24]. The U6 snRNA was used as the internal reference for miRNA, and BmGAPDH (Acc. MK243490) was used as the internal reference for mRNA in the present experiments.

### 2.7. Vector Construction and Luciferase Assay

MicroRNA mimics and inhibitors were synthesized by GenePharma (Shanghai, China). The wild-type and mutated luciferase reporter plasmid pmirGLO genes were constructed by inserting the CYP337A2 sequences between the firefly luciferase ORF and SV40 poly (A) into pmir-GLO vector. HEK293T cells were cultured in a 96-well plate maintained at 37 °C under 5% CO_2_ in Dulbecco’s modified Eagle’s medium (Thermo Fisher Scientific, Inc., Waltham, MA, USA) supplemented with 10% FBS, 1% penicillin/streptomycin mix, and its transfection with wildtype or mutated plasmids and miRNA mimics or negative control (NC) mimics, which was performed using the Calcium Phosphate Cell Transfection Kit (Beyotime). Each well contained 0.2 μg plasmid and 40 nmol/L miRNA mimics. One day after transfection, cells were lysed and subjected to luciferase assay using the Dual Luciferase Reporter Gene Assay Kit (Vazyme, Nanjing, China), according to the manufacturer’s instructions. Determination of the firefly and renilla luciferase activity was performed on SpectraMaxR i3 (Molecular Devices, Sunnyvale, CA, USA). The luciferase activity was normalized by comparison with the control groups. All experiments were performed in triplicate.

### 2.8. miRNA Mimics/Inhibitor Injection and Bioassays of Chlorfenapyr and Tralopyril

The third instar silkworm larvae were collected for microinjection by using microsyringes (Nanoliter 2000 Injector; WPI Inc., Sarasota, FL, USA). Each larva was injected with 1 μL of 100 μΜ Bmo-miR-6497-5P agomir (mimics) or antagomir (inhibitor), and the controls were injected with NC agomir or NC antagomir. qRT-PCR was used to measure the expression levels of miRNA and the target CYP450 gene. The procedure was the same as mentioned above. The reactions for all samples were run in triplicate with three biological replicates. The sensitivity of injected larvae to chlorfenapyr and tralopyril was determined following injection of miRNA agomir/antagomir. The injected larvae were treated with the approximate LC_30_ of chlorfenapyr (200 mg/L) and tralopyril (5 mg/L), respectively, using the leaf-dipping bioassays described above. Mortality was calculated 48 h after treatment. The tests were conducted in triplicate and each replicate contained 30 silkworm larvae.

### 2.9. Statistical Analyses

The LC_50_ and LC_30_ values and confidence limits were determined through regression based on the probit mortality concentration using the PROBIT procedure of SPSS16.0 software (SPSS Inc., Chicago, IL, USA). The statistical significance of gene expression, mortality, and luciferase activity was calculated using one-way ANOVA followed by Tukey’s multiple comparisons among different treatments. *p* < 0.05 was considered statistically significant.

## 3. Results

### 3.1. Susceptibility of Silkworm Exposed to Chlorfenapyr and Tralopyril

We evaluated the susceptibility of third instar *B. mori* larvae to chlorfenapyr and tralopyril to determine the LC_50_ and LC_30_ of these two chemicals (Table 1). After 48 h exposure, the estimated LC_50_ value of chlorfenapyr to the third instar larvae was 274.52 mg/L, while the LC_50_ value of tralopyril was only 8.24 mg/L. The estimated LC_30_ value of chlorfenapyr to the third instar larvae was 197.26 mg/L, which was much lower that of 4.94 mg/L of tralopyril. In addition, the estimated LC_10_ value of chlorfenapyr and tralopyril to the third instar larvae was 116.72 mg/L and 1.85 mg/L, respectively. These results strongly suggested that the third instar *B. mori* larvae were more sensitive to tralopyril than to chlorfenapyr.

### 3.2. Quantification of Chlorfenapyr and Tralopyril Residues in the Body

The retention time of chlorfenapyr and tralopyril was 10.06 and 8.76 min, respectively (Appendix A). At 0, 24, 48, and 72 h after feeding, the residues of chlorfenapyr in the larvae body were constant and maintained at higher levels when compared to tralopyril. At the beginning of the test, the content of tralopyril was not detected, and with the elapse of exposure time, it was increased but stayed in lower levels (Figure 1), indicating that chlorfenapyr can be metabolized into tralopyril in the silkworm bodies.

### 3.3. Illumina Sequencing and Sequence Alignment Analysis

A total of 33.47 to 40.44 M raw reads and 23.94 to 24.25 M clean reads were obtained from silkworm larvae treated with chlorfenapyr. A total of 30.08 to 32.10M raw reads and 23.92 to 24.71M clean reads were obtained from healthy silkworm larvae (Appendix A). The data were deposited in NCBI’s Sequence Read Archive (SRA) database with accession numbers SRR32005398, SRR32005397, SRR32005396, SRR32005395, SRR32005394, and SRR32005393. sRNAs with length ranging from 15 to 41 nt were screened from above clean reads, 23,936,496–24,249,871 sRNAs were obtained from larvae treated with chlorfenapyr, and 23,924,890 to 24,714,399 sRNAs were obtained from healthy silkworm larvae. There were relatively more sRNAs with the length ranging from 15 to 22nt in the larvae treated with chlorfenapyr and healthy silkworm larvae (Appendix A).

### 3.4. ncRNA Analysis

In this study, the ratio of total rRNA measured from each sample treated with chlorfenapyr ranged from 0.73% to 1.20%, and that of total reads measured from each sample of healthy silkworm ranged from 0.69% to 0.94%, both significantly lower than 40%. The results showed that the quality control index of RNA samples extracted was good in this study. The alignments of rRNA, tRNA, snRNA, Cis-reg, and other sequences with Rfam RNA databases in each treated or healthy sample were shown in Appendix A.

### 3.5. Identification of miRNAs

A total of 342 known miRNAs were obtained by comparing sequencing data with miRNA precursors in miRBase. In addition, after removing sRNAs, such as rRNA, tRNA, snRNA, and low-abundance reads, the secondary structure of other specific reads was predicted, but no corresponding matching results were found in miRbase. These specific reads with predicted secondary structure were considered as novel miRNAs, and those with the same seed sequence as known miRNAs of other species were grouped into conserved miRNAs. A total of 83 novel miRNAs were identified (Appendix A). The numbers of miRNA categories identified in each sample of the chlorfenapyr-treated group and the control group are shown in Appendix A.

### 3.6. Screening of Differentially Expressed miRNAs in Silkworm Larvae After Chlorfenapyr Treatment

Chlorfenapyr treatment significantly changed the expression of miRNA in silkworm larvae. A total of ten miRNAs were differentially expressed. Among these, six miRNAs were upregulated, and four miRNAs were downregulated (Table 2, Appendix A).

### 3.7. Predictions and Functional Annotation of the miRNA Target Genes

A total of 450 target genes of these 10 differentially expressed miRNAs were predicted. GO enrichment of miRNAs target genes was performed. Among these target genes, 431 were enriched into 46 GO subclasses under the three main clusters including “biological processes”, “cell components”, and “molecular functions”. Among these, the “biological process” contained the most subclasses (22), and the “cellular process” subclass contains the most target genes (193). Within the “cell components” clusters, “cell” and “cell part” subclasses contained the highest number of target genes (both 125), followed by “organelle” (85) and “organelle part” (58) subclasses (Appendix A). Within the “molecular function” cluster, the “binding” subclass contained the most target genes (93). In addition, a large number of target genes associated with detoxification were enriched in GO. For example, 83 target genes were enriched in “Metabolic processes” (Appendix A), indicating that chlorfenapyr treatment caused significant changes in the metabolic activity of silkworm larvae. KEGG enrichment analysis showed that 152 target genes were enriched in 27 subpathways, including 5 main pathways: cellular processes, environmental information processing, genetic information processing, metabolism, and biological systems. The categories with the highest number of target genes were “metabolism” and “organismal systems” (38 target genes), followed by “environmental information processing” (32 target genes), “genetic information processing” (24 target genes), and “cellular processes” (20 target genes). The subpathway with the highest numbers of target genes was “signal transduction” (26 target genes), followed by “folding, sorting and degradation” (18 target genes) (Appendix A). Other essential subpathways related to detoxification or metabolism were also enriched, such as “metabolism of terpenoids and polyketides” and “energy metabolism” subpathways (Appendix A).

### 3.8. Expression Patterns of Differentially Expressed miRNAs

Four differentially expressed miRNAs were selected for RT-qPCR verification for they were higher expressed according to the RNA-Seq result (with Log_2_FC value > 2, Table 1). In order to evaluate the effect of chlorfenapyr concentration on the expression level of miRNAs, we generated the low (120 mg/L for approximate LC_10_ value) and high (200 mg/L for approximate LC_30_) concentrations of chlorfenapyr. The results showed that the chlorfenapyr concentration and treatment duration had significant effects on the expression levels of these four miRNAs. The expression level of Bmo-miR-6497-5P was significantly increased at each time point after exposure to a high concentration of chlorfenapyr compared to control groups, while the expression level of Bmo-miR-6497-5p was significantly increased only after 48 h of low concentration chlorfenapyr treatment. After exposure for 24 h, there was no significant difference compared with the control group, and after treatment for 72 h, it was significantly downregulated compared with the control group (Figure 2). The expression level of Bmo-miR-6498-3P was significantly upregulated only after 24 h exposure to a low concentration of chlorfenapyr, while the expression level of Bmo-miR-6498-3P was significantly upregulated after 24 h and 72 h exposure to a high concentration of chlorfenapyr (Figure 2). In Bmo-miR-6498-5p, a low concentration of chlorfenapyr induced upregulation of its expression level only after 24 h of treatment, while a high concentration of chlorfenapyr caused upregulation of its expression level after 24 h and 72 h of treatment (Figure 2). Both low and high concentration of chlorfenapyr significantly induced upregulated expression of Bmo-miR-2999 after 72 h of treatment, but this upregulation expression pattern was not observed at the other two time points at either low or high concentration of chlorfenapyr treatment (Figure 2).

### 3.9. Screening of Differentially Expressed miRNA Targeting P450s

As Bmo-miR-6497-5P was upregulated at each time point after high concentration treatment with chlorfenapyr, we focused on the target P450 genes of Bmo-miR-6497-5P, and four P450 genes were obtained (Table 3). The binding region of Bmo-miR-6497-5P with the target genes is illustrated in Appendix A. Bmo-miR-6497-5P had the binding sites in the 3′ UTR or CDS region of all four P450 genes, and all the binding free energy was less than −20 kcal/mol (Appendix A). Similar to miRNAs’ expression levels, we also generated the lower (120 mg/L for approximate LC_10_ value) and higher (200 mg/L for approximate LC_30_) concentrations of chlorfenapyr. RT-qPCR was used to measure the expression patterns of these four P450 genes after chlorfenapyr treatment (Figure 3). The results showed that the expression level of *CYP337A2* was significantly downregulated compared with the control group at three time points both in low and high concentrations of chlorfenapyr, indicating that chlorfenapyr treatment significantly inhibited the expression level of *CYP337A2* (Figure 3). After either low or high concentration of chlorfenapyr exposure, the expression level of CYP333B1 was downregulated only at 24 h, while it was significantly upregulated at 48 h and 72 h (Figure 3). The expression level of CYP49A1 after 24 h and 48 h was significantly downregulated both in low or high concentration of chlorfenapyr exposure, while it was significantly upregulated in low concentration of chlorfenapyr treatment after 72 h (Figure 3). The expression level of CYP6AE3P was significantly upregulated under the high concentration of chlorfenapyr treatment after 72 h, in contrast, the two concentrations of chlorfenapyr did not cause this upregulated expression pattern after 24 h or 48 h (Figure 3).

### 3.10. Bmo-miR-6497-5P Regulates CYP337A2 Expression

As the expression level of *CYP337A2* was significantly downregulated at each time point after treatment with different concentrations of chlorfenapyr, which showed an opposite expression pattern to that of Bmo-miR-6497-5P, Bmo-miR-6497-5P and its target gene *CYP337A2* were selected as research objects in this study. In order to determine the targeted binding relationship between Bmo-miR-6497-5p and its candidate target gene *CYP337A2*, wild-type and mutant recombinant plasmid vectors were constructed using pmirGLO vectors for double luciferase reporter gene verification. The results showed that compared with the negative control (NC) of miRNAs’ mimics, the luciferase activity of Bmo-miR-6497-5p was significantly reduced after binding to the target gene *CYP337A2*. However, the luciferase activity of co-expressed *CYP337A2*-mut and Bmo-miR-6497-5p groups was not significantly reduced (Figure 4). Combined with the prediction of miRNAs and target sites (Figure 4), these results indicated that Bmo-miR-6497-5p can bind to *CYP337A2*, thus regulating the expression of *CYP337A2*.

To further confirm that *CYP337A2* is regulated by Bmo-miR-6497-5P, the third silkworm larvae were injected with the Bmo-miR-6497-5P agomir. The expression level of Bmo-miR-6497-5P was 2.9-fold greater in the larvae injected with Bmo-miR-6497-5P agomir than in the control group (i.e., the larvae injected with the NC agomir group) 24 h after injection (Figure 5A). In addition, the *CYP337A2* expression level in the larvae injected with the Bmo-miR-6497-5P agomir was significantly lower (Figure 5B). In contrast, the expression level of Bmo-miR-6497-5P in the larvae injected with Bmo-miR-6497-5P antagomir decreased by almost 50% compared to that in the NC antagomir group 24 h after injection (Figure 5C). Additionally, the expression level of *CYP337A2* in the silkworm larvae injected with Bmo-miR-6497-5P antagomir increased 3.8-fold (Figure 5D).

### 3.11. miRNA Mimics/Inhibitor Injection and Bioassays of Chlorfenapyr and Tralopyril

After injection of agomir, an agonist of Bmo-miRNA-6497-5p, and antagomir, an inhibitor of Bmo-miRNA-6497-5P, the toxicity of chlorfenapyr in the third instar larvae of silkworm was significantly changed. Injection of antagomir-miRNA significantly increased the mortality rate of silkworm larvae under chlorfenapyr treatment. In contrast, there were no significant differences in the mortality rate of silkworm larvae after the injection with agomir-miRNA, agomirNC, and antagomirNC (Figure 6). Regarding tralopyril treatment, there was no significant difference in mortality between the groups (Figure 6). In conclusion, after chlorfenapyr was digested by silkworm larvae, it may inhibit the expression of the target gene *CYP337A2* by upregulating the expression level of Bmo-miRNA-6497-5p. In this way, the bioconversion process of chlorfenapyr to tralopyril was alleviated, reducing the virulence of chlorfenapyr in silkworm larvae.

## 4. Discussion

Previous studies have demonstrated that the silkworm exhibits higher tolerance to chlorfenapyr compared to other insecticides [2,11]. However, the detailed molecular mechanism has not been discovered in depth. As chlorfenapyr must be transformed into toxic tralopyril and catalyzed by P450s and other enzymes, in order to perform functions in this form in insects [7,10], we validated the dynamics of chlorenapyr and tralopyril when silkworm larvae digested chlorfenapyr-coated mulberry leaves. The results showed that the content of chlofrenapyr was maintained at a high, constant level at each time point after feeding, while the content of tralopyril was not detected at the beginning of feeding and was kept in a relatively lower lever during the subsequent time points. These results strongly suggested that chlorfenapyr could be transformed into toxic tralopyril, but the transformation was significantly blocked in the silkworm. More importantly, it can be inferred there indeed exists a particular mechanism that was simpairing the process.

Chlorfenapyr stress and tralopyril stress have been shown to significantly upregulate the expression of 17 P450 genes in *Myzus persicae* [25]. By silencing the expression of *CYP6BG1*, its role in the resistance to chlorobenzamide in *Plutella xylostella* has been confirmed [26]. Similarly, the role of CYP6ER1, CYP302A1, and CYP3115A1 in the resistance of the brown planthopper *N. lugens* was evaluated by using RNAi methods. In particular, CYP6ER1 is involved in the resistance of *N. lugens* to thiamethoxam and dinotefuran, and knockout of *CYP6ER1* significantly enhanced the toxicity of thiamethoxam [27]. In the study of a non-target parasitoid wasp, *Meteorus pulchricornis*, after knocking down the P450 gene *CYP369B3*, the sensitivity of the wasp was significantly increased to phoxim, cypermethrin, and chlorfenapyr, respectively, thus proving that cytochrome P450s also play important roles in its pesticide tolerance in non-target insects [28]. In this study, three predicted target P450 genes were upregulated at certain time points after chlorfenapyr treatment in *B. mori* larvae, suggesting that these P450s may have functional roles in response to chlorfenapyr; as such, further study should be conducted to elucidate their exact roles in *B. mori.* Nevertheless, *CYP337A2* exhibits an interesting expression pattern in that it was downregulated at each time point after chlorfenapyr treatment, implying the possibility that *CYP337A2* was regulated by other factors.

As transcriptional regulators, microRNAs can regulate the expression of many target genes that are crucial in various physiological processes of plants and animals [29,30,31]. It has been confirmed that the generation and development of insecticide resistance are closely related to the regulation of target genes by microRNAs [32]. Li et al. (2015) found that miR-7a and miR-8519 could upregulate the expression of PxRyR in *Plutella xylostella*, thereby enhancing resistance to diamine insecticides [33]. Similarly, injection with miRNA-190-5p antagomir significantly increased *CYP6K2* abundance and thus improved chlorantraniliprole tolerance in *Spodoptera frugiperda* larvae [34]. In a recent study, Wang et al. (2023) discovered that the resistance of *Laodelphax striatellus* to triflumezopyrim was enhanced by attenuating the expression of miRNA PC-5P-30_205949, thereby activating the detoxification metabolic pathway by targeting CYP419A1 and ABCG23 [35]. Similarly, miR-278-3p is involved in the regulation of mosquito *Culex pipiens pallens* sensitivity to pyrethroids through decreased the expression of *CYP6AG11* [36]. Therefore, overexpression or depletion of miRNA can lead to resistance to insecticides.

In particular, overexpression of cytochrome P450 is associated with enhanced metabolic detoxification of insecticides [37,38,39]. For example, overexpression of *CYP6CY1* is involved in imidacloprid resistance in *Sitobion miscanthi* [40]. In contrast, recent studies have focused on whether the downregulated expression of cytochrome P450s is related to insecticide resistance or tolerance. One study found that the expression level of *CYP6N23* in DR-resistant C. pipiens was reduced by 3.6 times, and the mortality rate of sensitive strain was lower after the injection of dsCYP6N23. These results suggest that downregulating the expression of CYP6N23 promotes the development of pyrethroid resistance. Further double luciflucidase tests showed that miR-285 could inhibit the expression of CYP6N23 and ultimately reduce the mortality of Culex mosquitoes, leading to the speculation that miRNA-mediated P450 caused Culex mosquitoes to develop pyrethroid resistance [20]. In a recent study, Vandenhole et al. (2024) demonstrated that downregulation of P450 CYP392D8 activates chlorfenapyr to its active metabolite tralopyril, and they implied it was potentially one of the cases in which decreased activation as a likely resistance mechanism against chlorfenapyr [41].

In this study, the expression of the P450 gene *CYP337A2* in silkworm larvae was significantly downregulated after chlorfenapyr treatment. We then obtained differentially expressed miRNAs after chlorfenapyr treatment through differential transcriptome sequencing of sRNA and screened out Bmo-miR-6497-5P and its possible P450 target gene with the most significant upregulation. It is worth noting that the P450 gene predicted by miRNA target genes was different from the differentially expressed P450 gene after chlorfenapyr treatment reported in our previous study [2]. The reasons for this difference may be as follows: (1) The depth and methodology of differentially expressed gene sequencing were different from those of sRNA sequencing; (2) The two differential transcriptome sequencings were not performed in the same batch of chlorfenapyr-treated silkworm larvae. Among the four candidate P450 genes, the targeted regulatory relationship between Bmo-miR-6497-5P and *CYP337A2* was studied. The dual luciferase assay verified the targeting binding of Bmo-miR-6497-5P to *CYP337A2*. Since chlorfenapyr can be metabolized to toxic tralopyril by P450s in insects, the downregulated expression of P450 regulated by miRNA is an important reason for the inhibition of this transformation. In addition, the mortality of silkworm larvae exposed to chlorfenapyr and tralopyril was measured by injecting miRNA agonist agomir and inhibitor antago. The death rate was shown to increase after the injection of antago, a miRNA inhibitor. This strongly suggests that chlorfenapyr may be rapidly transformed into tralopyril after upregulation of *CYP337A2*, which was caused by downregulation of Bmo-miR-6497-5P, resulting in increased toxicity. A study on brown planthoppers reported that two newly identified miRNAs, mir-novel_85 and mir-novel_191, bound to the CYP6ER1 and CarE1 coding regions of the P450 gene in *N. lugens* and downregulated their expression, respectively. As such, injection of miRNA mimics and inhibitors significantly altered the sensitivity of *N. lugens* to nitenpyram [42]. To date, the results of studies have revealed that miRNAs can impair the mechanisms driving insecticide resistance by downregulating their target P450 genes in various insect pests [43,44,45,46]. In future studies, the detailed dynamics of transformation from chlorfenapyr to toxic tralopyril in the haemocoel of the silkworm larvae mediated by the association between miRNAs and their target P450s should be determined.

In conclusion, chlorfenapyr-treated silkworm larvae induced the upregulation of Bmo-miR-6497-5P and attenuated the expression of *CYP337A2*. It can be speculated that this interaction may lead to mitigation of the transformation from chlorfenapyr to toxic tralopyril, thus enhancing chlorfenapyr tolerance in silkworm larvae (Figure 7). This research is expected to contribute to further understanding of insecticide tolerance mechanisms in silkworm, in addition to the development of novel insecticides with higher selectivity used in mulberry pest management.

## Figures and Tables

**Figure 1 insects-16-00515-f001:**
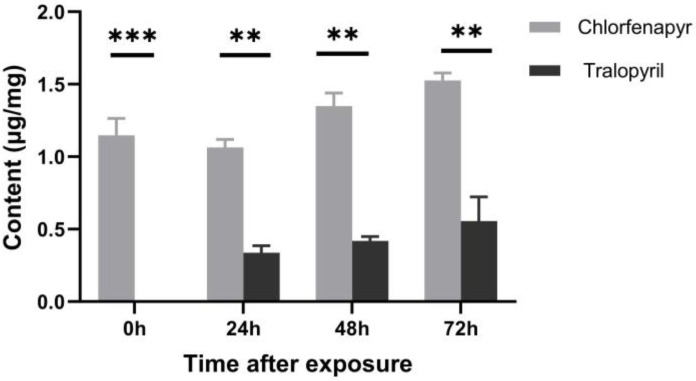
The content of chlorfenapyr and tralopyril in the silkworm larvae bodies after they digested chlorfenapyr-soaked mulberry leaves. The results were presented as means ± SE of three replicates. Differences in the content of each set of paired compounds were compared using a *t*-test. (** *p* < 0.01, *** *p* < 0.001).

**Figure 2 insects-16-00515-f002:**
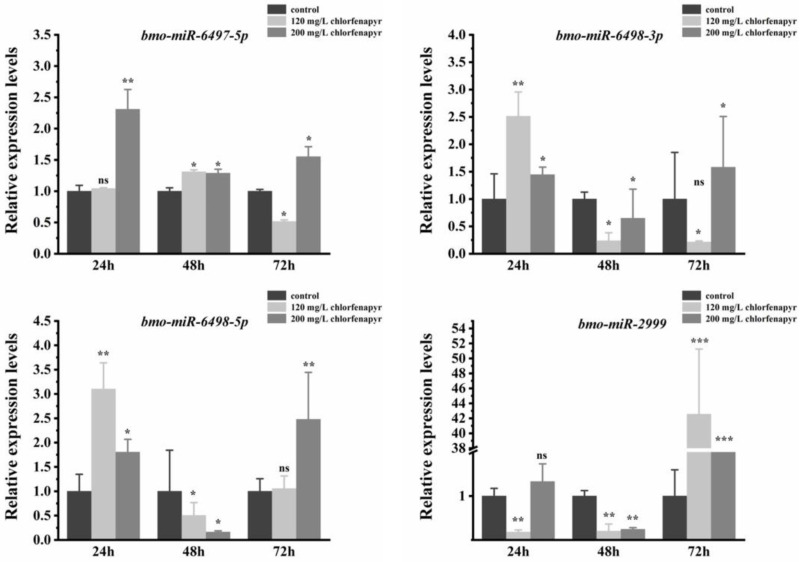
The expression patterns of Bm-miRNAs after the third silkworm larvae were exposed to chlorfenapyr. The results were presented as means ± SE of three replicates. Differences in the expression levels of each pair of miRNAs (insecticide-treated and control groups) were compared using a *t*-test. (* *p* < 0.05, ** *p* < 0.01, *** *p* < 0.001, ns: no significant differences).

**Figure 3 insects-16-00515-f003:**
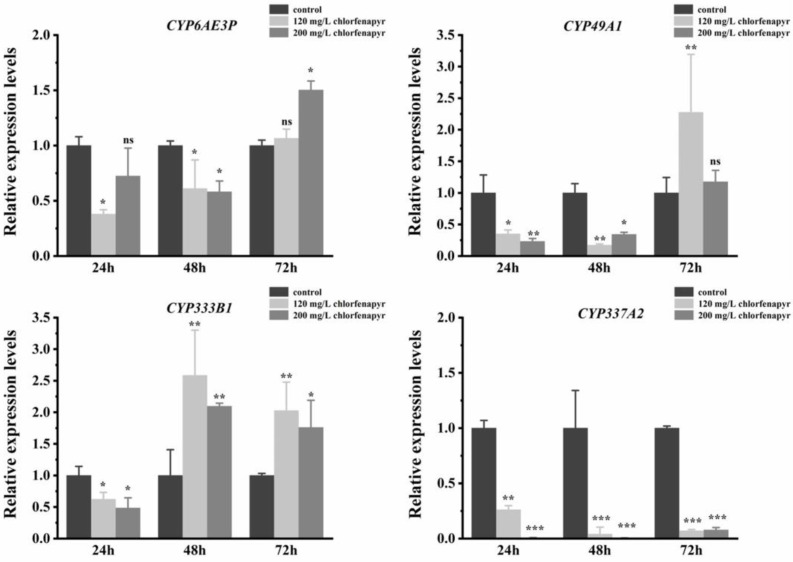
The expression patterns of P450 genes after the silkworm larvae were exposed to chlorfenapyr. The results were presented as means ± SE of three replicates. Differences in the expression levels of each pair of genes (insecticide-treated and control groups) were compared using a *t*-test. (* *p* < 0.05, ** *p* < 0.01, *** *p* < 0.001, ns: no significant differences).

**Figure 4 insects-16-00515-f004:**
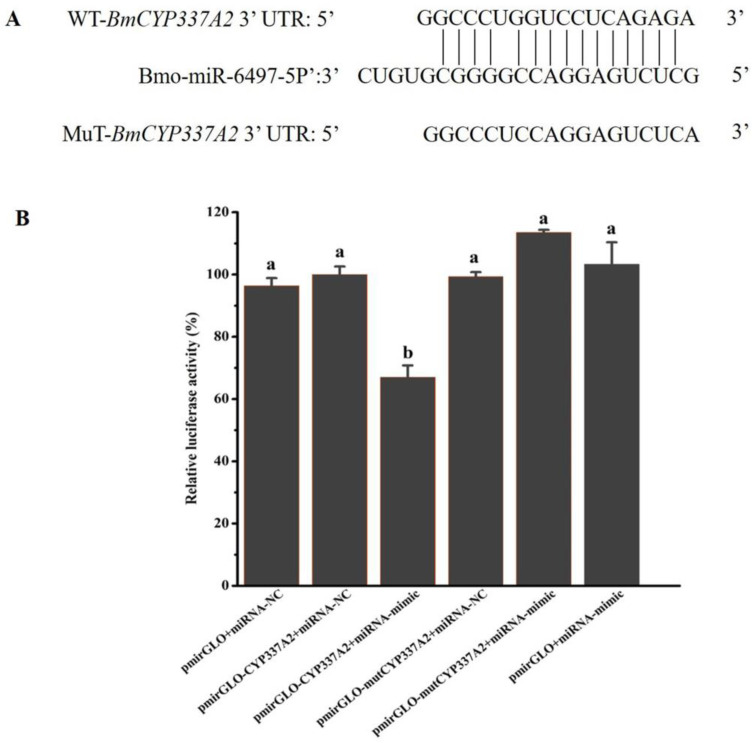
Interaction between Bmo-miRNA-6497-5P and CYP337A2. (**A**): The predicted target region of Bmo-miRNA-6497-5P located in the 3′ UTR of CYP337A2. (**B**): Double luciferase reporter assay. Different lowercase letters indicate significant differences based on a one-way ANOVA, followed by Tukey’s multiple comparison test (*p* < 0.05).

**Figure 5 insects-16-00515-f005:**
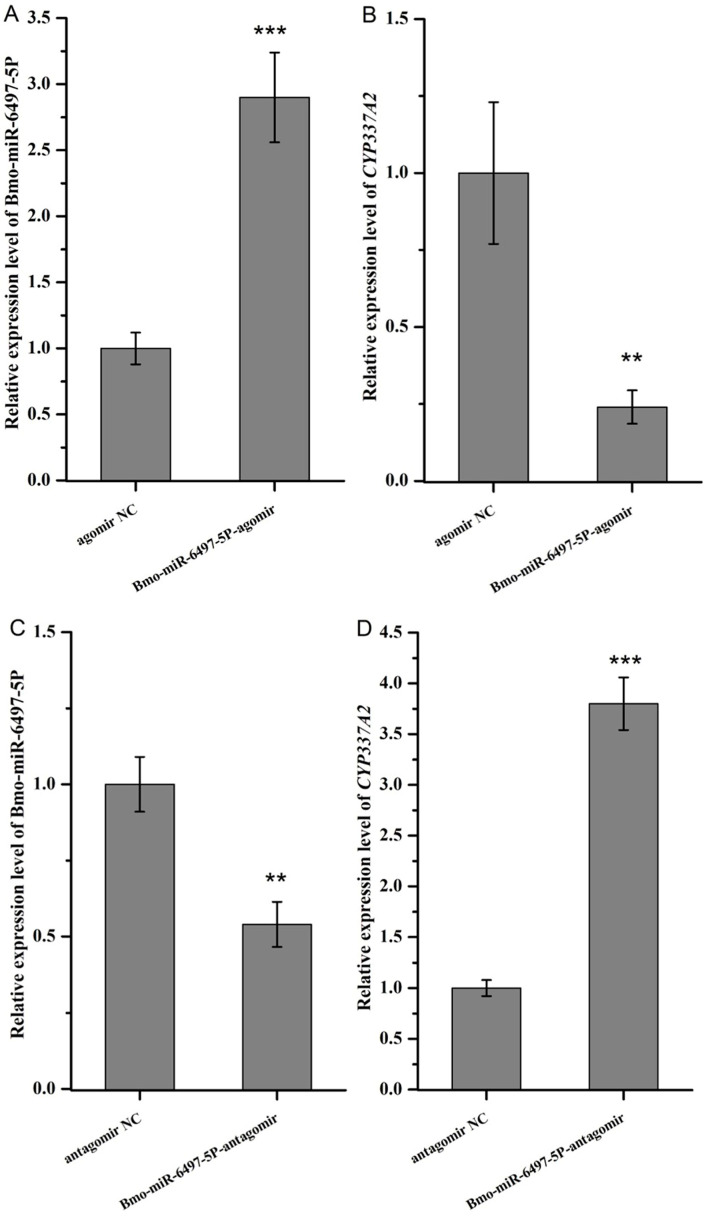
The impacts of Bmo-miRNA-6497-5P regulation on *CYP337A2* expression in *B. mori*. (**A**): Bmo-miRNA-6497-5P expression in *B. mori* larvae injected with Bmo-miRNA-6497-5P agomir. (**B**): *CYP337A2* expression in *B. mori* larvae injected with Bmo-miRNA-6497-5P agomir. (**C**): Bmo-miRNA-6497-5P expression in *B. mori* larvae injected with Bmo-miRNA-6497-5P antagomir. (**D**): CYP337A2 expression in *B. mori* larvae injected with Bmo-miRNA-6497-5P antagomir. (** *p* < 0.01, *** *p* < 0.001).

**Figure 6 insects-16-00515-f006:**
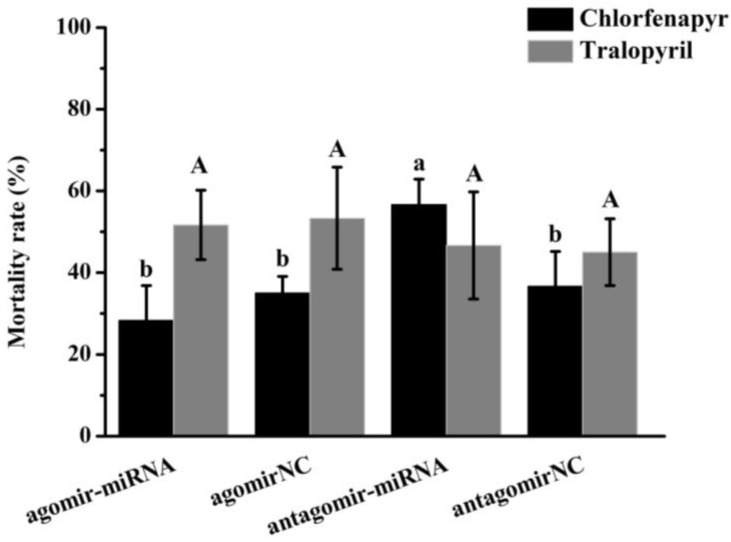
Mortality of silkworm larvae exposed to chlorfenapyr and tralopyril when they were injected with agomir and antagomir of Bmo-miRNA-6497-5p. ANOVA and TukeyHSD multiple comparison were used to compare the differences of mortality among treatments. (Differernt lower case letters indicate the significant differences among chlorfenapyr treatments, and the same upper case letters indicate there was no difference among tralopyril treatments, *p* < 0.05).

**Figure 7 insects-16-00515-f007:**
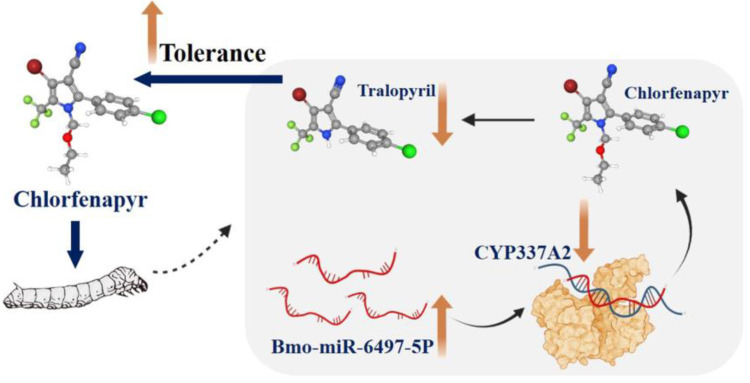
The schematic model of Bmo-miR-6497-5P regulates chlorfenapyr tolerance in silkworm larvae by targeting *CYP337A2*.

**Table 1 insects-16-00515-t001:** Susceptibility of *Bombyx mori* larvae to chlorfenapyr and tralopyril.

Chemicals	Slope ± SE	*χ* ^2^	*p* Value	LC_50_ (95% CI) * (mg/L)	LC_30_ (95% CI) * (mg/L)	LC_10_ (95% CI) * (mg/L)
Chlorfenapyr	5.38 ± 0.025	35.26	0.28	274.52 (215.42–342.55)	197.26 (129.16–241.03)	116.72 (97.35–125.54)
Tralopyril	2.23 ± 0.076	8.77	0.54	8.24 (5.38–13.87)	4.94 (0.29–10.25)	1.85 (0.92–2.34)

LC_50_ * = Lethal Concentration for 50%; LC_30_ * = Lethal Concentration for 30%; LC_10_ * = Lethal Concentration for 10%; CI = Confidence Interval.

**Table 2 insects-16-00515-t002:** Differentially expressed miRNAs between chlorfenapyr treatment and healthy *B. mori* larvae.

miRNAs	Sequence	Length	Log_2_FC	*p* Value	Regulation
Bmo-miR-275-5p	CGCGCTACTCCGGCGCCAGGACT	23	−1.02	2.77 × 10^−5^	Down
Bmo-miR-2792-3p	TAGATTAGATAGCGATTCCATT	22	−1.94	4.60 × 10^−2^	Down
Bmo-miR-2999	CTGCGACGGACTAGACGCGCA	21	2.18	1.22 × 10^−2^	Up
Bmo-miR-3000	CTGCGCTTAGATGAAGACACTA	22	1.52	3.02 × 10^−2^	Up
Bmo-miR-308-3p	AATCACAGGATAATACTGCGAG	22	1.12	1.12 × 10^−4^	Up
Bmo-miR-6497-5p	GCTCTGAGGACCGGGGCGTGTC	22	3.11	1.37 × 10^−3^	Up
Bmo-miR-6498-3p	AACGTCTGCGATGATACAGTT	21	4.27	1.00 × 10^−6^	Up
Bmo-miR-6498-5p	CGCGTCTGTTGTCGCAGCCGTGC	23	3.49	1.86 × 10^−3^	Up
Bmo-miR-79-5p	CTTTGGCGATTTAGCTCCGTGA	22	−1.10	3.91 × 10^−6^	Down
novel-47	CATGGTCTCATCATTCACA	19	−5.90	8.93 × 10^−5^	Down

**Table 3 insects-16-00515-t003:** P450 target genes of Bmo-miR-6497-5P.

miRNA	P450 Target Genes	Acc. Number in Gene Bank
Bmo-miR-6497-5P	*CYP337A2*	NM_001279385.1
Bmo-miR-6497-5P	*CYP333B1*	AK343202.1
Bmo-miR-6497-5P	*CYP6AE3P*	AK289290.1
Bmo-miR-6497-5P	*CYP49A1*	NM_001279490.1

## Data Availability

The original contributions presented in this study are included in the article/Appendix A. Further inquiries can be directed to the corresponding author.

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
