# Peer review of "microRNA Targeting Cytochrome P450 Is Involved in Chlorfenapyr Tolerance in the Silkworm, Bombyx mori (Lepidoptera: Bombycidae)"

_insects, 2025, doi:10.3390/insects16050515_

Round 1
Reviewer 1 Report
Comments and Suggestions for Authors
The manuscript microRNA targeting cytochrome P450 is involved in chlorfenapyr tolerance in silkworm, Bombyx mori". The topic is interesting enough to investigate, and the experiments are well-designed. The possible molecular mechanism of the tolerance to chlorfenapyr was discussed. However, some details need further discussion. The detailed comments are as follows:
- Line 19, “silkworm larvae after they were treated with”
- Line 20, supply the full name of CYP450, and the “CYP450” was the right abbreviation rather than “P450”.
- Line 23, 41, “dose”??? Concentration and dose have distinct meanings. Concentration refers to the relative amount in an environment or solution, while dose represents the absolute amount received by an organism. The author has confused these two terms in the manuscript, indicating a lack of understanding of their differences. It is recommended to carefully review and revise the manuscript to address this issue.
- Line 26-27, How can it be determined that it is CYP337a2 that is involved in the metabolism of chlorfenapyr? Any direct evidence? The results of qPCR and metabolism are only an indirect response to this.
- Line 39, supply the age of larvae.
- Line 42,44, “CYP337A2” should be italic. The names of the genes need to be italicized, so please check the full manuscript.
- Line 47, “pests” should be “insects”.
- Line 55, “long-term”
- Line 57, 103, “silkworms”
- Line 67, “the previous studies”.
- Line 71, “plays” an important
- Line 80, “insecticide” resistance
- Line 104, understanding “of” the molecular mechanism.
- Line 115, The reasons for 3rd larvae of the silkworm used for the test?
- Line 116, “2” Briefly
- Line 121, “air-dried”
- Line 133, “B. mori” should be italic
- Line 133, “larvae of B. mori were fed with mulberry leaves soaked with LC30 chlorfenapyr”. What is the amount of leaf feeding or the amount of the chlorfenapyr that has been calculated to be fed on the leaves? The amount of the chlorfenapyr affects the expression and metabolism of the gene. The reason for using LC30 for tests was?
- Line 144, “manufacturer’s protocol”
- Line 172, both “experiment” should be “experiments”
- Line 174, with the “parameters” as follows
- Line 175 “demanding” strict 5’ seed pairing
- Line 182, “was” should be “were”
- Line 226, the “P” should be italic, please check the manuscript.
- In 3.1, the valve of LC30 should be provided, because this concentration was used for silkworm exposure.
- Line 240, the content of tralopyril was
- The font in Figure 1 was different from other figures.
- Line 283-284, this should be in 2.9 Statistical analyses, also the version of these softwares should be provided.
- Line 309, ten miRNAs were differentially expressed, but why were only four genes selected here? Any explanation?
- In Figures 2, 3, the reasons for the usage of concentrations (120 and 200 mg/L), why these key information was not provided in 2. Materials and Methods.
- In Figure 2, the error line is too long, indicating poor repeatability. Why was the expression of bmo-miR-2999 significantly upregulated in the 120 mg/L treatment group, any explanation? This selection criterion is a bit unreasonable,
- Line 335, “Due to Bmo-miR-6497-5P was up-regulated at each time point after high concentration treatment of chlorfenapyr, the target P450 genes of Bmo-miR-6497-5P were focused on”. The expression levels of these four mRNAs change with treatment time, which is a normal phenomenon and may also be a mechanism for insects to cope with stress. We cannot assume that Bmo-miR-6497-5P is the focus of research just because its expression level increases at high concentrations; At the same time, the expression level of this mRNA decreased in the low concentration treatment group at 72 hours, while it increased in the high concentration treatment group. What may be the possible reasons?
- In Figure 3, Is there a coordinated relationship between the expression profiles of four CYP genes regulated by the same miRNA under pesticide stress? Perhaps certain genes work in a low expression state? This requires purifying the corresponding CYP450 enzyme to determine which CYP450 gene/enzyme is responsible.
- Line 383, “larvae were injected with the Bmo-miR-6497-5P agomir”. How long after the injection is the qPCR test conducted?
- Line 400, In experiments 3.10 and 3.11, how many silkworms have been exposed? exposure time? The concentration of exposure? These key pieces of information are missing.
- Line 440 and knocking out of
- Line 449 CYP337A2 “exhibits” an interesting expression pattern.
- Line 454 and the development of “insecticide” resistance.
- Line 504, the “detailed” dynamics of transformation from chlorfenapyr.
Author Response
Reviewer 1
The manuscript microRNA targeting cytochrome P450 is involved in chlorfenapyr tolerance in silkworm, Bombyx mori". The topic is interesting enough to investigate, and the experiments are well-designed. The possible molecular mechanism of the tolerance to chlorfenapyr was discussed. However, some details need further discussion. The detailed comments are as follows:
Line 19, “silkworm larvae after they were treated with”
Response: Response: We accept this comment and revised it in the text.
Line 20, supply the full name of CYP450, and the “CYP450” was the right abbreviation rather than “P450”.
Response: We accept this comment and revised it.
Line 23, 41, “dose”??? Concentration and dose have distinct meanings. Concentration refers to the relative amount in an environment or solution, while dose represents the absolute amount received by an organism. The author has confused these two terms in the manuscript, indicating a lack of understanding of their differences. It is recommended to carefully review and revise the manuscript to address this issue.
Response: We greatly appreciate and accept this comment, and carefully review and revise the manuscript.
Line 26-27, How can it be determined that it is CYP337a2 that is involved in the metabolism of chlorfenapyr? Any direct evidence? The results of qPCR and metabolism are only an indirect response to this.
Response: We greatly appreciate these comments. In the present study, we used the HPLC analysis to determine the metabolism of chlorfenapyr when it was digested by silkworm larvae and the results showed that it can be biotransformed into tralopyril. Then, the qPCR results showed that CYP337A2 and Bmo-miR-6497-5p exhibiting opposite expression patterns after chlorfenapyr treatment. Further luciferase activity measurement indicated that Bmo-miR-6497-5p can bound to CYP337A2. Meanwhile, miRNA mimics/inhibitor injection and bioassays of chlorfenapyr and tralopyril suggested that the mortality of silkworm larvae injected with the antagomir of Bmo-miR-6497-5p was significantly increased after exposure to sublethal concentration chlorfenapyr. These results strongly implied miRNA may bind to its target CYP450 gene to increase the tolerance to chlorfenapyr. More detailed investigation, such as HPLC analysis that determine the content of chlorfenapyr and tralopyril when the silkworm larvae were injected with miRNA mimics/inhibitor, will be conducted in future studies. To avoid too absolute conclusion, we change this sentence to “Based on these results, it is possible that Bmo-miR-6497-5p may target CYP337A2, regulating its expression and silkworm increases its chlorfenapyr tolerance by upregulating Bmo-miR-6497-5p expression, thereby inhibiting the biotransformation of chlorfenapyr to toxic tralopyril catalysed by CYP337A2.” in the revised text.
Line 39, supply the age of larvae.
Response: We appreciate this comment and supply the age of larvae in the text (3rd instar larvae).
Line 42,44, “CYP337A2” should be italic. The names of the genes need to be italicized, so please check the full manuscript.
Response: We appreciate and accept this comment. We revise the full manuscript and make all the gene names to be italicized.
Line 47, “pests” should be “insects”.
Response: We accept this comment and revise it in the text.
Line 55, “long-term”
Response: We accept this comment and revise it in the text.
Line 57, 103, “silkworms”
Response: We accept this comment and revise them in the text.
Line 67, “the previous studies”.
Response: We accept this comment and modified it in the text.
Line 71, “plays” an important
Response: We accept this comment and revise it in the text.
Line 80, “insecticide” resistance
Response: We accept this comment and revise it in the text.
Line 104, understanding “of” the molecular mechanism.
Response: We accept this comment and revise it in the text.
Line 115, The reasons for 3rd larvae of the silkworm used for the test?
Response: We appreciate this comment. Indeed, when concerning the bioassays of insecticides towards silkworm, although various instars can be used, 3rd instar is the main candidate (Xie et al., 2018; Zhang et al., 2019; Dai et al., 2019). In the present study, the body size of 3rd instar silkworm larvae is moderate and if we select higher instar, e.g. 4th or 5th, silkworm larvae are approach spinning and pupation, and the change of inner physiology of silkworm are momentous. Due to the tolerance or detoxication to insecticides also correlates with inner physiology, in order to avoid these interference, 3rd instar was selected.
References:
Xie, D.Y., Yand, Z.G., Chai, J.P., Tian, M.J., Bai, X.R., Zhang, Y.H., Luo, Y.J. Determination of Chronic Toxicity of Eight Pesticides to Silkworm. Agrochemicals, 2018, 57(6): 438-442.
Dai, J.Z., Chen, W.G., Yang, Y.P., Lin, W.H., Sun, H.Y., Qian, Q.J. Toxicity Evaluation of 13 Kinds of Pyrethroid Insecticides to Bombyx mori. Science of Sericulture, 2019, 45(4): 0610-0613.
Zhang, F., Yang, Y.P., Chen, W.G., Qian, Q.J., Lin, W.H., Dai, J.Z., Sun, H.Y. Toxicity Evaluation of Imidacloprid to Silkworm Bombyx mori. Bulletin of Sericulture, 2019, 50(2): 8-10.
Line 116, “2” Briefly
Response: We appreciate this comment and delete “2” in the text.
Line 121, “air-dried”
Response: We accept this comment and revise it in the text.
Line 133, “B. mori” should be italic
Response: We accept this comment and revise it in the text.
Line 133, “larvae of B. mori were fed with mulberry leaves soaked with LC30 chlorfenapyr”. What is the amount of leaf feeding or the amount of the chlorfenapyr that has been calculated to be fed on the leaves? The amount of the chlorfenapyr affects the expression and metabolism of the gene. The reason for using LC30 for tests was?
Response: We greatly appreciate this comment. When we conducted the HPLC analysis, the leaf-dipping bioassays was used. Briefly, mulberry leaf discs (7 cm diameter) were cut and dipped into the insecticide solutions for 30 s and air dried for 30 min at room temperature. The silkworm larvae were fed with these discs and most of the discs had been digested by them. Therefore, it was certain that chlorfenapyr entering the body of silkworm larvae together with mulberry leaves. The purpose of HPLC analysis was to investigate the dynamics of chlorfenapyr and whether it can be biotransformed into tralopyril. Although it cannot measure the exact amount of chlorfenapyr, the result of this test can answer that chlorfenapyr would be metabolized to tralopyril. The reason for using LC30 for tests was that LC30 was the sublethal concentration that could survive some of tested silkworm and the subsequent measurement, such as HPLC analysis, can be conducted.
Line 144, “manufacturer’s protocol”
Response: We accept this comment and revise it in the text.
Line 172, both “experiment” should be “experiments”
Response: We accept this comment and revise it in the text.
Line 174, with the “parameters” as follows
Response: We accept this comment and revise it in the text.
Line 175 “demanding” strict 5’ seed pairing
Response: We accept this comment and revise it in the text.
Line 182, “was” should be “were”
Response: We accept this comment and revise it in the text.
Line 226, the “P” should be italic, please check the manuscript.
Response: We accept this comment and revise it in the text.
In 3.1, the valve of LC30 should be provided, because this concentration was used for silkworm exposure.
Response: We appreciate and accept this comment. The valve of LC30 has been provided in the revised text.
Line 240, the content of tralopyril was
Response: We accept this comment and revise it in the text.
The font in Figure 1 was different from other figures.
Response: We appreciate and accept this comment. Figure 1 has been modified in the revised version.
Line 283-284, this should be in 2.9 Statistical analyses, also the version of these softwares should be provided.
Response: We appreciate this comment. This sentence has been addressed in 2.5 Bioinformatic analysis section and we delete it here in the revised text. Besides, the version of software has been provided.
Line 309, ten miRNAs were differentially expressed, but why were only four genes selected here? Any explanation?
Response: We greatly appreciate this comment. The criterion for selecting these four miRNAs is that they were higher expressed in the RNA-Seq result compared to other miRNAs (Log2FC value > 2). In the future study, we will focus on other six miRNAs for the possible novel findings. We also address this criterion in the revised text.
In Figures 2, 3, the reasons for the usage of concentrations (120 and 200 mg/L), why these key information was not provided in 2. Materials and Methods.
Response: We appreciate and accept this comment. We are sorry for this omitting. the LC10 value of chlorfenapyr to the third instar larvae was 116.72 mg/L, and LC30 value was 197.26 mg/L, in order to prepare the insecticide solutions conveniently, approximate integer value was selected and generated the lower (120mg/L for LC10) and higher (200mg/L for LC30) concentrations. We address this key information in the materials and methods sections and results sections.
In Figure 2, the error line is too long, indicating poor repeatability. Why was the expression of bmo-miR-2999 significantly upregulated in the 120 mg/L treatment group, any explanation? This selection criterion is a bit unreasonable.
Response: We appreciate this comment. Indeed, the error line is a bit long in some of the bars, and it may be due to the variations of the inner physiology of different biological samples that they were treated with chlorfenapyr. Despite this, please note the break in the y-axis which represents the upper and lower limits are far higher than one, indicating the expression levels bmo-miR-2999 was much higher in low concentration chlorfenapyr treated groups than control cohorts. Besides, bmo-miR-2999 significantly upregulated in the 120 mg/L, as well as in the 200mg/L treatment, suggesting bmo-miR-2999 was significantly induced at 72h timepoint. The selection criterion for miRNA and its target CYP450 gene is mainly based on their expression levels after chlorfenapyr treatment, in which exhibited the opposite expression patterns. Although bmo-miR-2999 was significantly upregulated at 72h timepoint, the expression in other two timepoints was downregulated or comparative regardless chlorfenapyr concentrations. on the contrary, bmo-miR-6497-5p was upregulated at all three timepoints except for the 72h after low concentration treatment. This implied bmo-miR-6497-5p maybe the ideal candidate miRNA for further investigation.
Line 335, “Due to Bmo-miR-6497-5P was up-regulated at each time point after high concentration treatment of chlorfenapyr, the target P450 genes of Bmo-miR-6497-5P were focused on”. The expression levels of these four mRNAs change with treatment time, which is a normal phenomenon and may also be a mechanism for insects to cope with stress. We cannot assume that Bmo-miR-6497-5P is the focus of research just because its expression level increases at high concentrations; At the same time, the expression level of this mRNA decreased in the low concentration treatment group at 72 hours, while it increased in the high concentration treatment group. What may be the possible reasons?
Response: We greatly appreciate these comments. Indeed, it is a normal phenomenon that the expression levels of miRNAs changed with treatment time, the expression level of Bmo-miR-6497-5P was quite different from other three miRNAs after chlorfenapyr treatment that it was up-regulated at each time point only except in the low concentration treatment group at 72 hours. This up-regulating expression patterns can help us and give cues to conduct the subsequent investigation, in other words, to narrow the range of miRNA selection, and we successfully screened Bmo-miR-6497-5P and its target CYP450 CYP337A2. Meanwhile, it cannot neglect the possible roles of other miRNAs in the chlorfenapyr tolerance and we will conduct further investigation for them in the future studies. It should be noticed that Bmo-miR-6497-5P was down-regulated in the low concentration treatment group at 72 hours, while it increased in the high concentration treatment group. The possible reasons for this may because that the expression patterns of miRNAs rely on the concentration of chlorfenapyr after a longer treatment time, in other words, higher concentration of chlorfenapyr can result in the action of miRNA rather than low concentration. Alternatively, another possible explanation was that low concentration of chlorfenapyr after longer treatment time could induce other miRNA to mediate the tolerance mechanism which requires further validation.
In Figure 3, Is there a coordinated relationship between the expression profiles of four CYP genes regulated by the same miRNA under pesticide stress? Perhaps certain genes work in a low expression state? This requires purifying the corresponding CYP450 enzyme to determine which CYP450 gene/enzyme is responsible.
Response: We greatly appreciate these comments. Indeed, it is possible that there is coordinated relationship between these four CYP genes, regulating by the same miRNA under chlorfenapyr stress. Although the results of luciferase activity test and the bioassay after miRNA mimics/inhibitor injection implied that CYP337A2 was the potential target gene, we cannot deny that certain genes work in a low expression state. The view that purifying the corresponding CYP450 enzyme to determine the responsible CYP450 gene/enzyme is very useful and constructive, however, it would take a much longer period to obtain the detail results and undoubtedly, we would conduct this work in our future studies. Many thanks again for this suggestion.
Line 383, “larvae were injected with the Bmo-miR-6497-5P agomir”. How long after the injection is the qPCR test conducted?
Response: We appreciate this comment. The qRT-PCR test was conducted 24h after agomir or antagomir injection. We have added this information in the revised version.
Line 400, In experiments 3.10 and 3.11, how many silkworms have been exposed? exposure time? The concentration of exposure? These key pieces of information are missing.
Response: We appreciate and accept this comment. We are sorry for the missing information. In experiment 3.10 for the expression levels of miRNAs and target CYP450 gene, the agomir and antagomir injection groups and the NC agomir or NC antagomir were run in triplicate with three biological replicates. In experiment 3.11, mortality was calculated after 48 h of exposure. The concentration of exposure was approximate LC30 of chlorfenapyr (200 mg/L) and tralopyril (5mg/L). The tests were conducted in triplicate and each replicate contains 30 silkworm larvae. We have provided the information in the materials and methods section.
Line 440 and knocking out of
Response: We accept this comment and revise it in the text.
Line 449 CYP337A2 “exhibits” an interesting expression pattern.
Response: We accept this comment and revise it in the text.
Line 454 and the development of “insecticide” resistance.
Response: We accept this comment and revise it in the text.
Line 504, the “detailed” dynamics of transformation from chlorfenapyr.
Response: We accept this comment and revise it in the text.
Reviewer 2 Report
Comments and Suggestions for Authors
Dear Authors,
1) For the title, scientific names (without scientific authority) should be followed by the order and family placement
2) It is recommended to include the abstract, an overview of the methods, details of the treatments or assessments, key results presented with their corresponding values and statistical significance, and a conclusion summarizing the evaluation or interpretation of the experimental findings.
3) Delete 2 in line 116 and insert a reference for this technique.
4) Please briefly give leaf-dipping technique - volume of dipping solution. Size and weight of leaves. Were leaves cut to be of same size and weight? Were these weighed pre and post-feeding? Control?
5) B. mor in all manuscript must be italic.
6) Why did the authors choose the selected genes in Table 1?
7) The resolution of Fig. 2 needs to improve.
8) The figs 4 and 5 are very small.
9) I recommended transferring Table S2 to the manuscript.
10) The authors should focus on presenting your work by discussing the results step by step, while relocating some of the citations from the Discussion section to the Introduction, where they are more appropriately placed
11) Needs extensive English editing
Comments on the Quality of English LanguageNeeds extensive English editing
Author Response
Dear Authors,
1) For the title, scientific names (without scientific authority) should be followed by the order and family placement
Response: We accept this comment and revise it in the text.
2) It is recommended to include the abstract, an overview of the methods, details of the treatments or assessments, key results presented with their corresponding values and statistical significance, and a conclusion summarizing the evaluation or interpretation of the experimental findings.
Response: We appreciate and accept this comment. We have re-organized the Abstract section as the reviewer’s comment.
3) Delete 2 in line 116 and insert a reference for this technique.
Response: We accept this comment and revise it in the text.
4) Please briefly give leaf-dipping technique - volume of dipping solution. Size and weight of leaves. Were leaves cut to be of same size and weight? Were these weighed pre and post-feeding? Control?
Response: We greatlly appreciate this comment. The leaf-dipping bioassays was conducted to determine the susceptibility of 3rd instar B. mori larvae to chlorfenapyr and tralopyril, respectively. Briefly, mulberry leaf discs which were cut to the same size with 7 cm diameter were dipped into the insecticide solutions (with the volume of 200 ml) for 30 seconds and air-dried for 30 minutes at room temperature. The silkworm larvae were fed with these discs individually and most of the discs had been digested by them when we checked the mortality. Therefore, it was certain that insecticides entering the body of silkworm larvae together with mulberry leaves. The leaf discs were not weighed pre and post-feeding in insecticides treatment and control groups.
5) B. mor in all manuscript must be italic.
Response: We accept this comment and revise it in the full text.
6) Why did the authors choose the selected genes in Table 1?
Response: We appreciate this comment. The criterion for selecting these four miRNAs is that they were higher expressed in the RNA-Seq result compared to other miRNAs (Log2FC value > 2). In the future study, we will focus on other six miRNAs for the possible novel findings. We also address this criterion in the revised text.
7) The resolution of Fig. 2 needs to improve.
Response: We appreciate and accept this comment. The resolution of Fig. 2 has been improved.
8) The figs 4 and 5 are very small.
Response: We appreciate and accept this comment. Figs 4 and 5 have been enlarged in the revised version.
9) I recommended transferring Table S2 to the manuscript.
Response: We appreciate and accept this comment. Table S2 has been transferred to the main manuscript marked as Table 1 in the revised version.
10) The authors should focus on presenting your work by discussing the results step by step, while relocating some of the citations from the Discussion section to the Introduction, where they are more appropriately placed
Response: We appreciate and accept this comment. The Discussion section has been re-organized carefully in the revised version.
11) Needs extensive English editing
Response: We appreciate and accept this comment. We have used the English editing service from MDPI Author Services for the extensive the English editing.
Round 2
Reviewer 1 Report
Comments and Suggestions for Authors
The revision well addressed my previous concerns.
Reviewer 2 Report
Comments and Suggestions for Authors
I highly recommend accepting this manuscript in the present form